# Optimal sizing and power losses reduction of photovoltaic systems using PSO and LCL filters

**Mohammed F. Elnaggar[1,2], Armel Duvalier Péné[3,4], André Boussaibo[3], Fabrice Tsegaing[3], Alain Foutche Tchouli[5], Kitmo[6]\*, Fabé Idrissa Barro[4]**

**1** Department of Electrical Engineering, College of Engineering, Prince Sattam Bin Abdulaziz University, Al-Kharj, Saudi Arabia, **2** Department of Electrical Power and Machines Engineering, Faculty of Engineering, Helwan University, Helwan, Egypt, **3** Department of Electrical Engineering, University Institute of Technology, University of Ngaoundere, Ngaoundere, Cameroon, **4** Physics Department, Laboratory of Semiconductors and Solar Energy, Faculty of Science and Technology, University Cheikh Anta Diop, Dakar, Senegal, **5** Department of Electrical Engineering Energetic and Automatic, Laboratory of Energy, Signal, Images, Automatic (LESIA), National Advanced School of Agro-Industrial Sciences, University of Ngaoundere, Ngaoundere, Cameroon, **6** Department of Renewable Energy, National Advanced School of Engineering of Maroua, University of Maroua, Maroua, Cameroon

\* kitmobahn@gmail.com

**Data Availability Statement:** All relevant data are within the paper and its supporting information files.

**Funding:** The author(s) received no specific funding for this work.

## Abstract

The integration of renewable energy systems into electricity grids is a solution for strengthening electricity distribution networks (SEDNs). Renewable energies such as solar photovoltaics are suitable for reinforcing a low-voltage line by offering an electrical energy storage system. However, the integration of photovoltaic systems can lead to problems of harmonic distortion due to the presence of direct current or non-linear feedback in networks from other sources. Therefore, connection standards exist to ensure the quality of the energy before injection at a point of common coupling (PCC). In this work, particle swarm optimization (PSO) is used to control a boost converter and to evaluate the power losses and the harmonic distortion rate. The test on the IEEE 14 bus standard makes it possible to determine the allocation or integration nodes for other sources such as biomass, wind or hydrogen generators, in order to limit the impact of harmonic disturbances (LIHs). The evaluation of the harmonic distortion rate, the power losses as well as the determination of the system size is done using an objective function defined based on the integration and optimization constraints of the system. The proposed model performs better since the grid current and voltage are stabilized in phase after the photovoltaic source is injected.

## 1. Introduction

With the increasing production of air pollution today, the sustainable use of electrical energy remains fundamental to saving the planet [1]. Fossil fuels are dangerous because of environmental problems such as global warming and air pollution [2]. That's why generating electrical power from renewable energy sources is the ideal way to protect the universe [3]. There are

**Competing interests:** The authors have declared that no competing interests exist.

several renewable energy sources, but photovoltaic generators are the most widely recognized as one of the most important and offering strong possibilities for easy and less costly integration [4]. Unfortunately, integrating photovoltaic generators into the distribution system requires a good knowledge of the systems involved in shaping electrical parameters [5]. Many system parameters need to be taken into account, such as the harmonic currents injected by the grid, power quality, islanding and the ability to provide auxiliary services in adverse conditions. These standards guarantee the interconnection of photovoltaic systems in electricity grids [6]. The best-known standard is that of the Institute of Electrical and Electronics Engineers (IEEE) and the International Electrotechnical Commission (IEC). Other national standards also exist in most countries and need to be taken into account. The installation of renewable energy systems is subject to a number of compliance tests [7]. A feasibility assessment must first be carried out and verified. For this reason, photovoltaic generators must undergo extensive testing. Several types of test system have been developed [8]. One of these is called a PV emulator. This allows photovoltaic panels to be replaced and the photovoltaic inverter to be tested easily, taking into account different conditions such as solar irradiation and temperature. The role of this emulator model is to stimulate the characteristics of the photovoltaic generator. This controlled source consists in emulating the different I-V and PV characteristics of any photovoltaic field, taking into account specific ambient factors. Two types of controlled voltage source are generally considered: the linear power amplifier and the switching power converter. Unfortunately, these emulators are used on PV models that can be more or less complex. This requires knowledge of more advanced algorithms. For example, one aspect not taken into account in these emulators is a model with a skewed region of non-linear dynamic characteristics, both forwards and backwards. As far as the linear power amplifier is concerned, its dynamic performance is quite high. Given its low efficiency, it is not suitable for medium- to high-power applications.

Some emulators [9] have been developed to study the connections of distributed generators, but they have the disadvantage of being limited to conventional electrical systems (such as protective relays, industrial equipment and transformers), wind turbines and wave energy converters. Several types of network emulator have been proposed, such as the linear power amplifier, the transformer emulator and the switching power amplifier [10]. However, these systems have specific requirements that are not associated with other electrical system components, so they cannot be used.

The most widely used test is the IEEE Standard [11], which evaluates the allocation and sizing of distributed generations. In this article, the proposed system is implemented and modeled in Matlab/Simulink. The parameters of the various nodes present on a radial system are obtained using the Newton Raphson method.

The integration of photovoltaic systems into distribution networks can affect their efficiency, especially in transient and stable modes. For this reason, the injection of these systems into the grid remains a major challenge in modern grid developments to date, and their connection is likely to present undesirable effects that require special attention [12]. These include harmonic distortion, power factor disturbance, DC injection and island operation. Generally speaking, grid-connected photovoltaic systems are considered ideal if we have: continuous operation, where they work around nominal values, reduced peaks in electrical parameters, the possibility of automatic disconnection or connection in the event of failure [13].

The aim of this work is to contribute to the improvement of electrical power quality in a grid-connected PV system. As the PV system is considered as decentralized generation, national grid connection codes and injection standards have to be taken into account. To achieve this objective, an LCL Filter is used to shape the signals from a multi-level inverter. The PSO is used to determine the size of the overall system, as well as to locate possible

connections to other complementary sources in the event of distribution network reinforcement. In addition, an objective function is used to evaluate system performance and energy control and management at the common point of coupling.

This work is structured in three parts: An introduction reviewing the state of the art, materials and methods presenting the mathematical tools used to formalize the overall system, results and discussion devoted to the analysis and interpretation of the data, and conjecture following the initial observations and hypotheses. Finally, a conclusion justifies the work's contribution and outlines future prospects.

## 2. Materials and methods

### 2.1 System topology and configuration

In order to give an overall representation of the system under consideration in this article, a photovoltaic generator is depicted in Fig 1. The DC converter is used to transfer energy to a DC bus where DC loads can be connected. A battery bank is connected to the DC bus. At the output of the DC bus, an inverter provides AC voltages for filtering using the LCL filter. The inverter voltages and currents are collected at a point of common coupling from which several loads can be coupled. From this point, the electrical network and the loads from the household

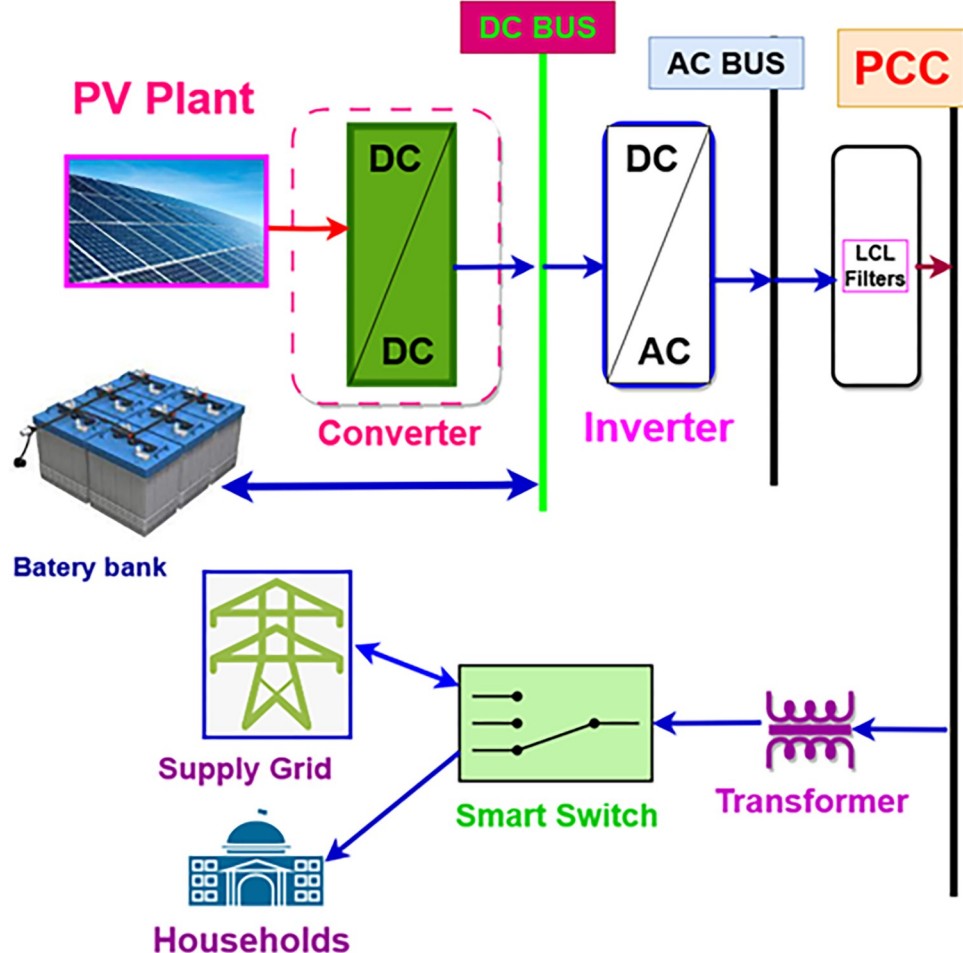

**Fig 1. System configuration at common point of coupling.**

are supplied through a conditioning circuit. The control of the energy delivered or subscribed to the networks and the energy delivered to the households is ensured using an intelligent switch. A unidirectional transformer enables the required voltage level to be supplied downstream of the networks and to households.

## 3. System based photovoltaic plant

The photovoltaic energy system is based on the efficiency of the solar panels [14]. This system includes module parameters such as irradiance and temperature under standard conditions, defined by Eqs (1), (2) and (3).

$$P_{PVG}(t) = \eta_{PVG} \times A_{PVG} \times G_s(t) \tag{1}$$

$$\eta_{PVG} = \eta_{STC} \times \eta_{MPP}[1 - \tau(T_C - T_{STC})] \tag{2}$$

$$T_C = T_a + \left[\frac{NOCT - 20}{800}\right] \times G_{PVG}(t) \tag{3}$$

## 4. Evaluation of power loss on IEEE bus

Fig 2 depicts the configuration of the standard IEEE 14 bus test with radial configuration, which enabled the various nodes where a source can be injected or a supply can be provided to be located [15]. The Fig 2 illustrates the various connection nodes with associated voltage and current levels. At each of these nodes, it is possible to depict the electrical parameters [16] in accordance with Eq (4).

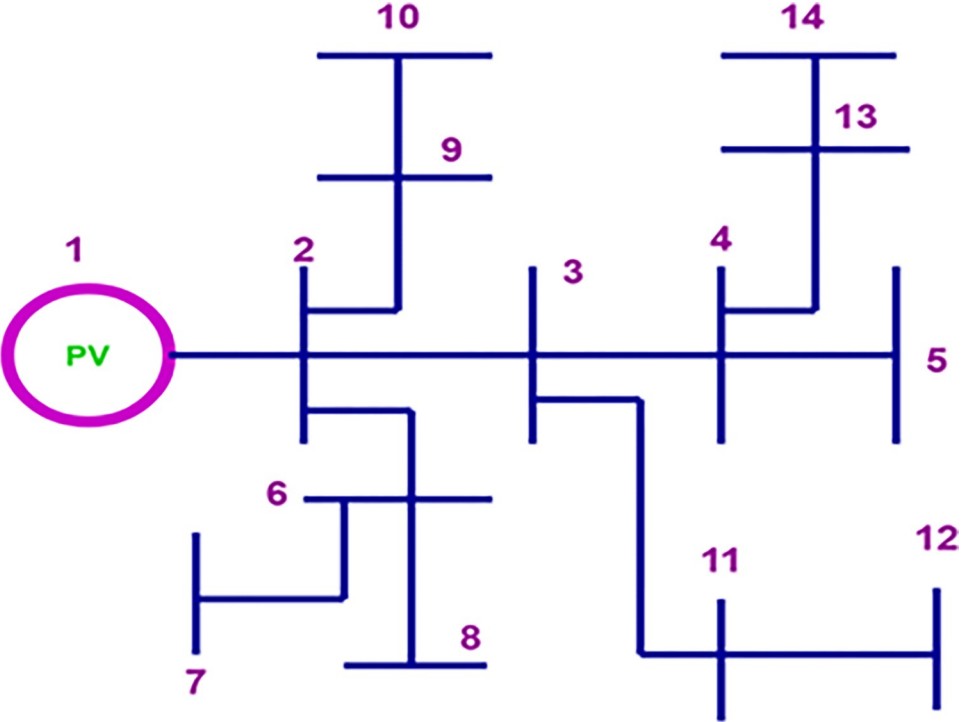

**Fig 2. System configuration on IEEE 14-bus.**

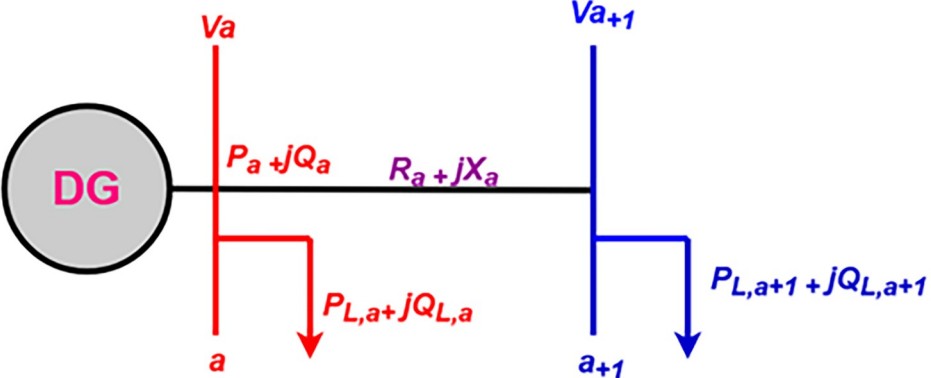

**Fig 3. Bus line representation.**

The proposed system consists of a photovoltaic generator, which is the only distributed generation. Fig 3 is a model of a portion of a line between two points A and B of the system of a three-phase system [17]. The mathematical model of this line is given by Eqs (4) and (5). These equations are used to implement the constraints on power loss reduction [18] that are considered by the objective function in Eq (15).

The assessment of the power losses is done on a line between two points A and B by the following eqs. At these points correspond the line voltages and currents, defined as [19]:

$$V_b = V_a + I_a(\theta_a - j\mu_a) \tag{4}$$

Where $V_b = V_{a+1}$

$$I_a = \left(\frac{P_{3a} + jQ_{3x}}{V_a}\right)^* \tag{5}$$

$$Pl_{oss(a,b)} = (P_a{}^2 + Q_a{}^2) * \theta_a \tag{6}$$

$$I_a = \left(\frac{P_a{}^2 + Q_a{}^2}{|V_a|}\right)^* * \mu_a \tag{7}$$

$$P_{Tloss} = \sum_{i=1}^{n} P_{loss(a,b)} \tag{8}$$

## 5. Particles swarm optimization

The PSO particle swarm optimization [20] is one of the best algorithms for the optimal search of a system. Compared to the genetic algorithm which is better suited for complex problems and whose speed of convergence is slow compared to that of PSO and fast Some algorithms such as perturb and observe(P&O) do not react well to sudden changes in input variables. Others are limited by the size of the system to be controlled. The PSO was developed by [21]. It is based on the position of particles with different velocities. The optimum of the solutions is determined for the best scores obtained thanks to a cost function. The position of these particles is randomly generated according to Eq (9). The parameters and constraints are given by Eqs (11) and (12). For: $X_m$, the position, $V_i$, velocity, Pbest best position and $C_1$ and $C_2$

weighing coefficients. Here values are: $C_1 = C_1 = 1.2$; $\omega_{\max} = 0.8$, $\omega_{\min} = 0.3$.

$$X_m = (x_{m,1}, x_{m,2}, x_{m,3}, L, x_{m,n}) \tag{9}$$

$$S_{j\psi}^{i+1} = S_{j\psi}^{i} + v_{j\psi}^{i+1}, k = 1, 2, \cdots, n; \psi = 1, 2, \cdots, m \tag{10}$$

$$S_{kd}^{i+1} = \omega v_{kd}^{i} + c_1 * rand(pbest_{kd} - s_{kd}^{i}) + c_2 * rand(gbest_{kd} - s_{kd}^{i}) \tag{11}$$

$$\omega_f = \omega_{\max} - \frac{\omega_{\max} - \omega_{\min}}{f_{\max}} f \tag{12}$$

$$Pbest_a = (pbest_{a,1}, pbest_{a,2}, L, pbest_{a,b}) \tag{13}$$

$$Gbest_a = (gbest_{a,1}, gbest_{a,2}, ....gbest_{a,b}) \tag{14}$$

## 5.1 Objective function

$$Funct = Fitness1 + Fitness2 \tag{15}$$

$$\begin{cases} Fitness1 = \min(P_{loss}) = \sum_{a=1}^{n} I_a^2 \theta_a \\ Fitness2 = \min f(N_{PV}, N_{Batt}, E_{ESS}) \end{cases} \tag{16}$$

*Constraints*

$$\begin{cases} 0 \le P_{DG} \le \sum P_{load} \\ 0 \le Q_{DG} \le \sum Q_{load} \\ V_{\min} \le V_i \le V_{\max} \end{cases} \tag{17}$$

Where $V_{\min}$ and $V_{\max}$ are respectively minimum at maximum voltage at bus.
$i = 1,2,3,\cdots,N$ and $I_i \le I_i^\chi$

## 5.2 Configuration of the system PSO algorithms

The flowchart of the PSO algorithm used for power management at the common coupling point and maximum power extraction at the DC/DC converter is given in Fig 4. The input parameters are the nominal voltages and power, the energy produced by the system and the constraints on the rate of harmonic distortion. This flowchart includes the voltages at the various nodes of the IEEE 33 bus.

## 6. Filter LCL topology and configuration

Harmonic suppression is achieved by LCL filters as depicted in Fig 5. The output voltage of the multilevel inverter has a sinusoidal waveform. The shape of the inverter output voltage waveform shows, after calculating the rate of harmonic distortion, that it is not full of harmonics. The LCL filter is used to remove harmonics with higher frequencies and of odd rank compared with the fundamental frequency. The grid current and voltage are synchronized. The equivalent diagram of the filter is depicted in Fig 5.

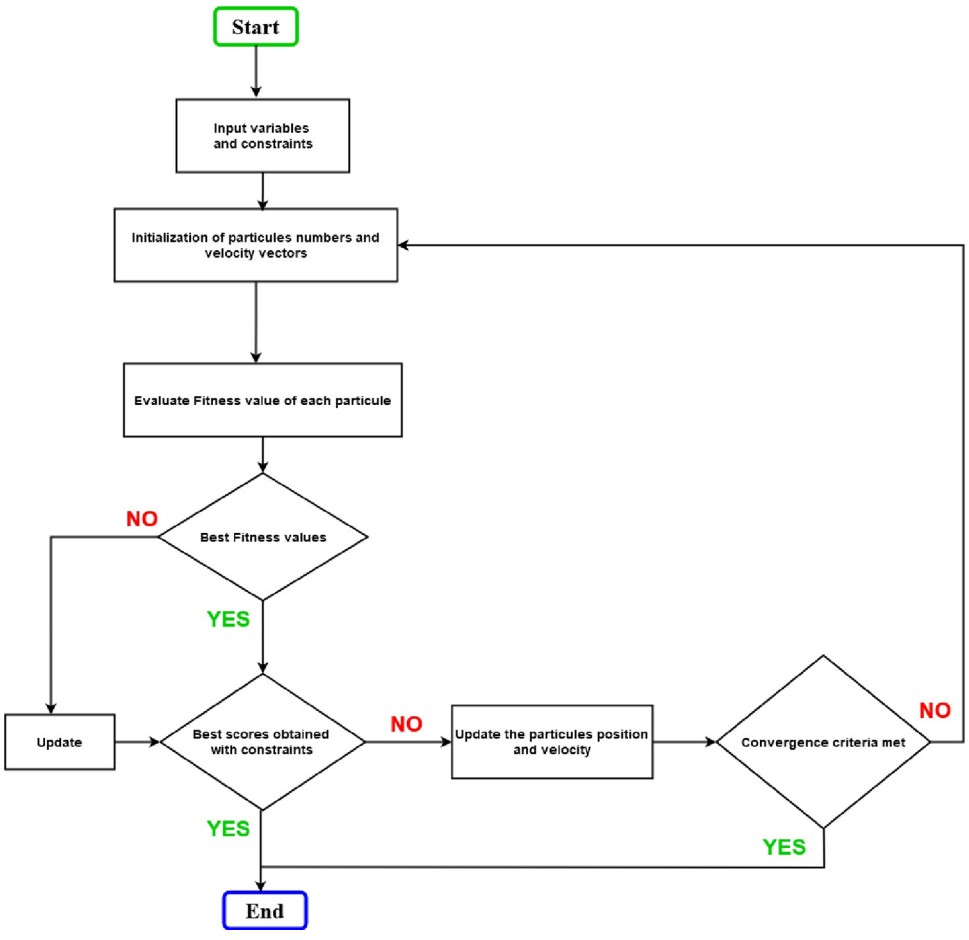

**Fig 4. PSO algorithms flowchart.**

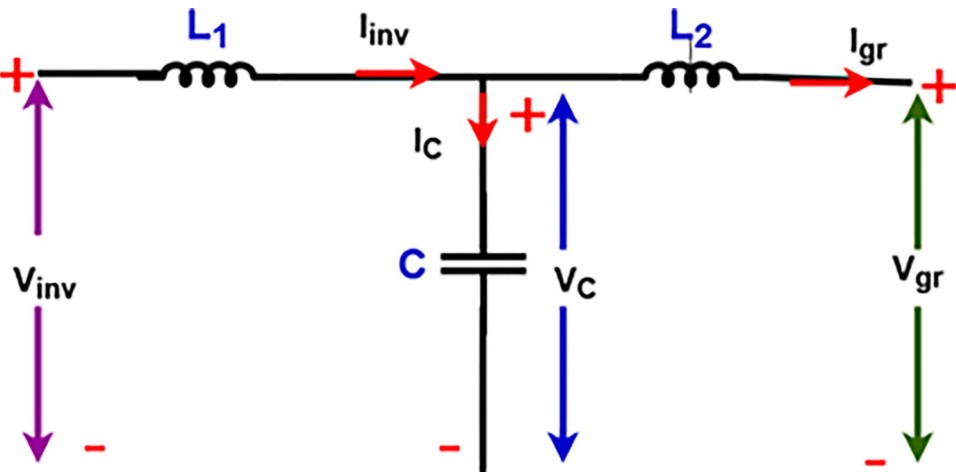

**Fig 5. LCL filter topology.**

## 7. Inverter and LCL filter grid connection

The inverter is controlled in duty cycle and modulated using PWM. The LCL filter connected to its output acts as a low-pass filter to suppress higher-order harmonics. For this purpose, the LCL filter injects two complex poles conjugated around the cut-off frequency Fres, given by following Eqs (Eqs 18, 19 and 20).

$$\frac{di_{gr}}{dt} = \frac{1}{L_2}v_c - \frac{1}{L_2}v_{gr} \tag{18}$$

$$\frac{di_{inv}}{dt} = \frac{1}{L_1}v_{inv} - \frac{1}{L_1}v_c \tag{19}$$

$$\frac{dv_c}{dt} = \frac{1}{C}i_{inv} - \frac{1}{C}i_{gr} \tag{20}$$

$$x_1 = i_{gr} \tag{21}$$

$$x_2 = \dot{x_1} = \frac{di_{gr}}{dt} \tag{22}$$

$$x_3 = \dot{x_2} = \frac{d^2 i_{gr}}{dt^2} \tag{23}$$

$$\dot{x_2} = \frac{d^2 i_{gr}}{dt^2} = \frac{1}{L_2}\left[\frac{1}{C}i_{inv} - \frac{1}{C}i_{gr}\right] - \frac{1}{L_2}\dot{Vgr} \tag{24}$$

$$\dot{x_3} = \frac{d^3 i_{gr}}{dt^3} = \frac{1}{L_2 C}\left[\frac{1}{L_1}v_{inv} - \frac{1}{L_1}v_c\right] - \frac{1}{L_2 C}\left[\frac{1}{L_2}v_c - \frac{1}{L_2}v_{gr}\right] - \frac{1}{L_2}\ddot{Vgr} \tag{25}$$

$$\begin{pmatrix} \dot{x_1} \\ \dot{x_2} \\ \dot{x_3} \end{pmatrix} = \begin{pmatrix} 0 & 1 & 0 \\ 0 & 0 & 1 \\ 0 & -\omega_o^2 & 0 \end{pmatrix} \begin{pmatrix} x_1 \\ x_2 \\ x_3 \end{pmatrix} + \begin{pmatrix} 0 \\ 0 \\ \varepsilon \cos \omega t \end{pmatrix} \tag{26}$$

$$H_{LCL} = \frac{1}{s^3 C L_1 L_2 + s(L_1 + L_2)} \tag{27}$$

The transfer function which defines the LCL filter is given by Eq (28), with an associated damping resistor (Rd).

$$G_{LCL}(s) = \frac{R_d C s + 1}{L_1 L_2 C s^3 + (L_1 + L_2)C R_d s^2 + (L_1 + L_2)s} \tag{28}$$

$$F_{res} = \sqrt{\frac{L_i + L_g}{L_i L_g C f}} \tag{29}$$

## 8. Evaluation of the power quality

The rate of total harmonic distortion (THD) is an important parameter for injecting energy at the common point of coupling. The IEEE 519–2014 standard specifies the different values to be applied. The THD in voltage and the THD in current are respectively given in Eqs (30 and 31). In these equations, $V_1$ represents the fundamental voltage and $I_1$, the current corresponding to the fundamental voltage. $V_h$ is the root mean square voltage and $I_h$, the root mean square current, n is the harmonic number.

$$THD_V = 100 * \frac{\sqrt{\sum_{h=2}^{n} V_h^2}}{V_1} \tag{30}$$

$$THD_I = 100 * \frac{\sqrt{\sum_{h=2}^{n} I_h^2}}{I_1} \tag{31}$$

## 9. Results and discussion

### 9.1 Voltage profile of DC converter

Power extraction using Maximum Power Point Tracking (MPPT) methods, based on Particle Swarm Optimization (PSO), offers an excellent output voltage stability performance for static DC converters. These methods are important for identifying the best scores for a photovoltaic system. Fig 6, illustrates the boost converter output voltage. This voltage is obtained and stabilized using alpha duty cycle control. It can be seen that this type of converter is suitable for stabilizing the DC voltage at around 96V. This demonstrates that in the event of disturbances, when there is no shading effect on the solar modules, these algorithms have a significant impact in stabilizing the DC voltage level.

### 9.2 Currents profile of the inverter

Fig 7 depicts the instability of the system at the common point of coupling for a short duration of disturbances due to the presence of non-linear loads in the system. It can be observed that

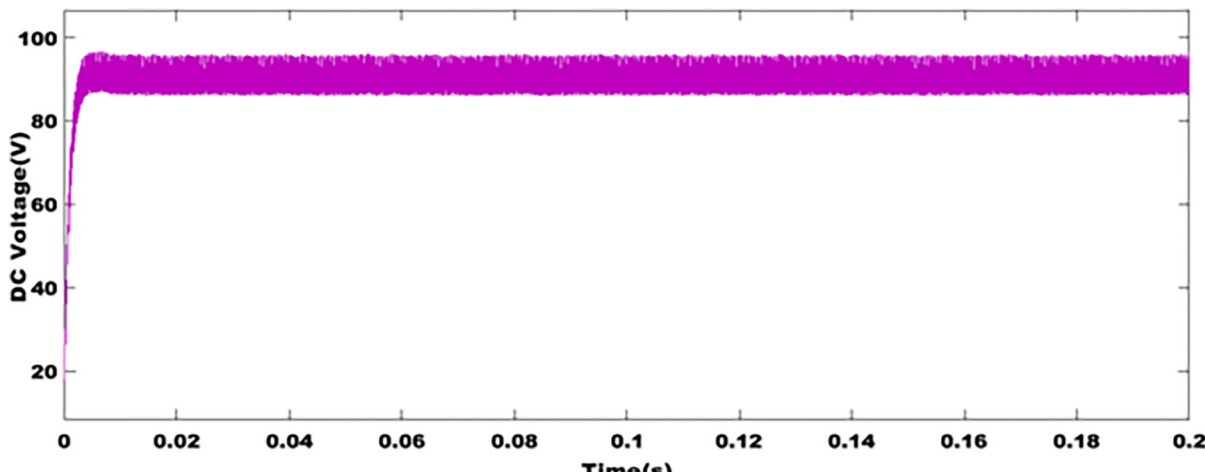

**Fig 6. DC voltage profile.**

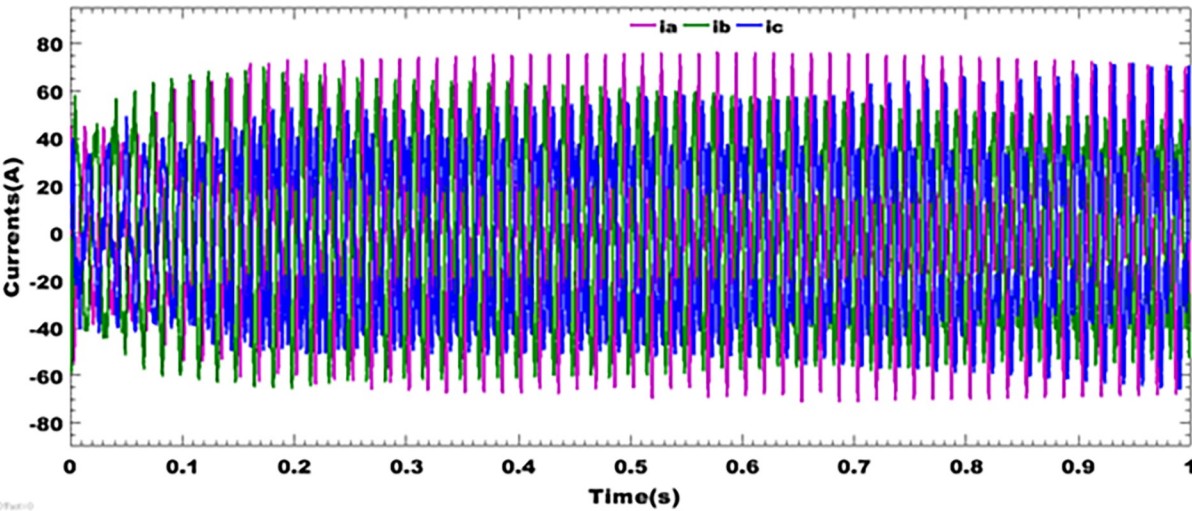

**Fig 7. Three phase currents of the inverter.**

for a short duration transient, the proposed system exhibits current peaks in the absence of an electrical energy storage system. This shows that for all three phases of the currents at the common coupling point, harmonics are present even in the presence of filters. It implies that, in the absence of an energy storage system, the use of LCL filters does not guarantee the best quality of energy from a photovoltaic system for injection into a centralized power grid.

## 9.3 Profile of load currents

Without the LCL filters, the inverter output currents are full of harmonics and offer no advantage for linear loads as shown in Fig 8. Three-phase currents have peaks with a large number of odd-numbered harmonics that are multiples of three. This type of current can be used to power asynchronous machines used in conventional industrial drive systems. The use of such machines offers no advantage or guarantee in terms of the lifetime of the installation or

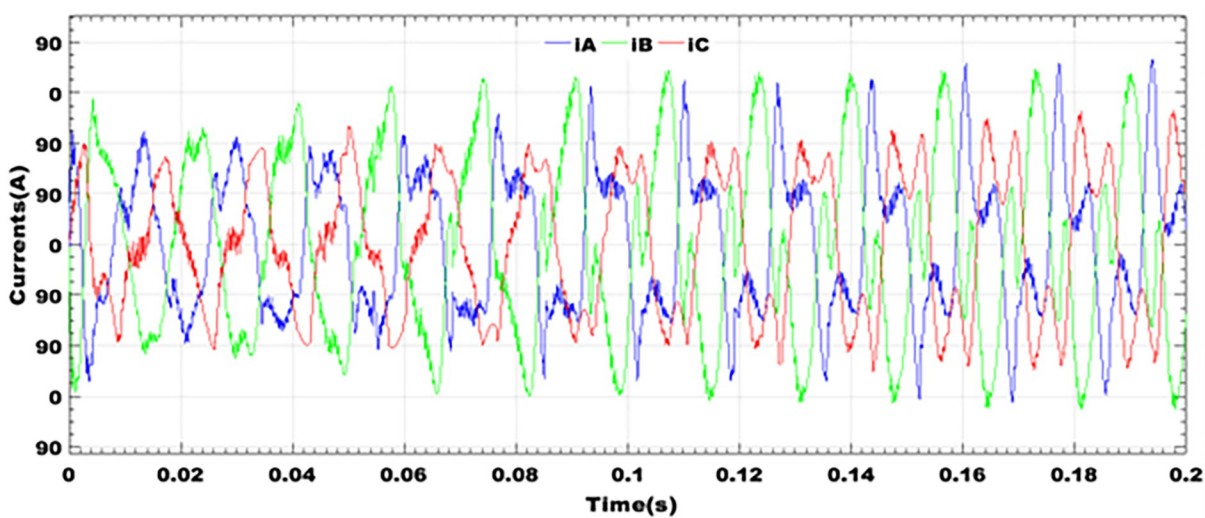

**Fig 8. Profile of load currents without filter.**

continuity of service in electrical distribution networks. These methods therefore constitute a discouragement to decentralized production, whereas the policy of self-consumption should be encouraged in order to achieve green cities.

### 9.4 Harmonics currents profiles

Fig 9 depicts the superposition of harmonic current and network current, per network phase. It can be seen that the harmonic currents are very high, and this has an impact on the load current. The characteristics of this harmonic current illustrate that, in the absence of filters, this system is unsuitable for grid integration. In the absence of an LCL filter, and in the case of high current demand or non-linear loads, there is a loss of synchronism. These transient disturbances over long periods of time can lead to deterioration of the installation on the one hand, and on the other hand, lead to a reduction in the system's service life. And when an electrical installation system becomes obsolete before the depreciation period has expired, economic losses are incurred. The aim of distributed generation is to amortize production and operating costs, before expecting to make a profit.

### 9.5 Grid voltages profiles during disturbances

Fig 10 depicts the three-phase voltages at the output of the multilevel inverter. These voltages are alternating and $2\pi/3$ out of phase from each other. This figure explains how power grid disturbances in the event of a non-linear load have a negative impact on the inverter's output voltage when the photovoltaic system is connected to the distribution grid. Based on this result, it can be seen that three-phase voltages are balanced around 2 kV. This voltage level may not be sufficient when current demand is very high. This is why an electrical energy storage system is essential.

### 9.6 Grid currents profiles during blackout

Fig 11 depicts the currents of three network phases under the effects of harmonic disturbances due to the presence of non-linear loads and sudden variation. The Fig 11 depicts troughs and peaks in the presence of repeated blackouts, especially for a duration of 0.45 seconds and for a duration of 0.78 seconds. This current dip problem stems from the fact that when the battery

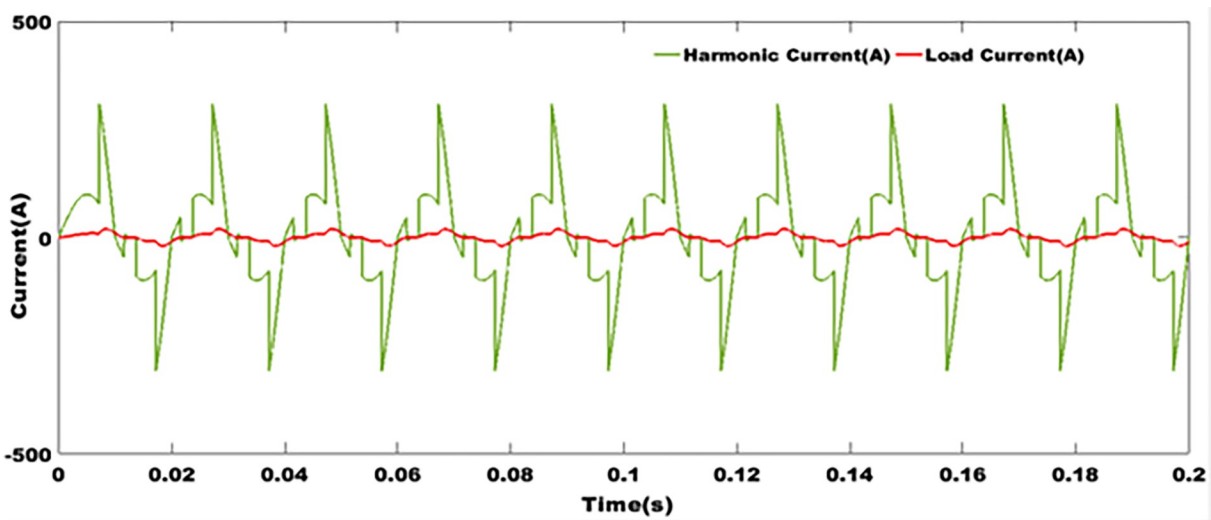

**Fig 9. Harmonics currents profiles.**

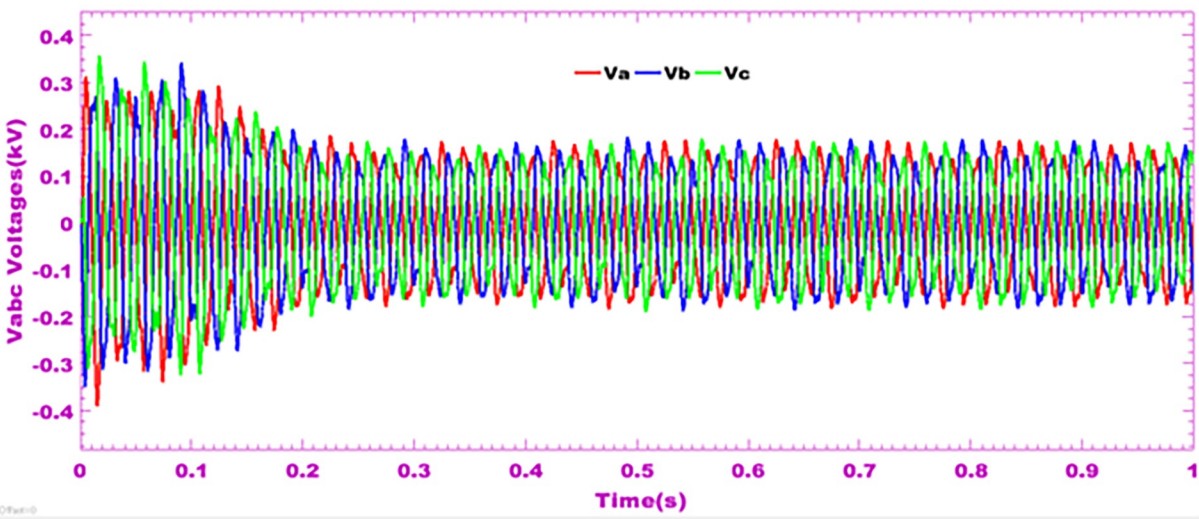

**Fig 10. Grid voltages profiles during disturbances.**

bank is not connected, the system is highly faulty and cannot be considered for interconnection to other primary sources. This result corroborates the fact that the energy storage system is always associated with the photovoltaic generator, even when these are connected to the grid. On the other hand, the electrical energy storage system is much more widely used for isolated sites. So, when current demand is very high, it's essential to integrate battery banks, even for a system connected to the grid in urban areas.

### 9.7 Load currents and voltages after filtering

Fig 12 depicts the output voltage at the household level for a high demand from another part of the system in the absence of the battery banks. Thus, not only is the current attenuated, but

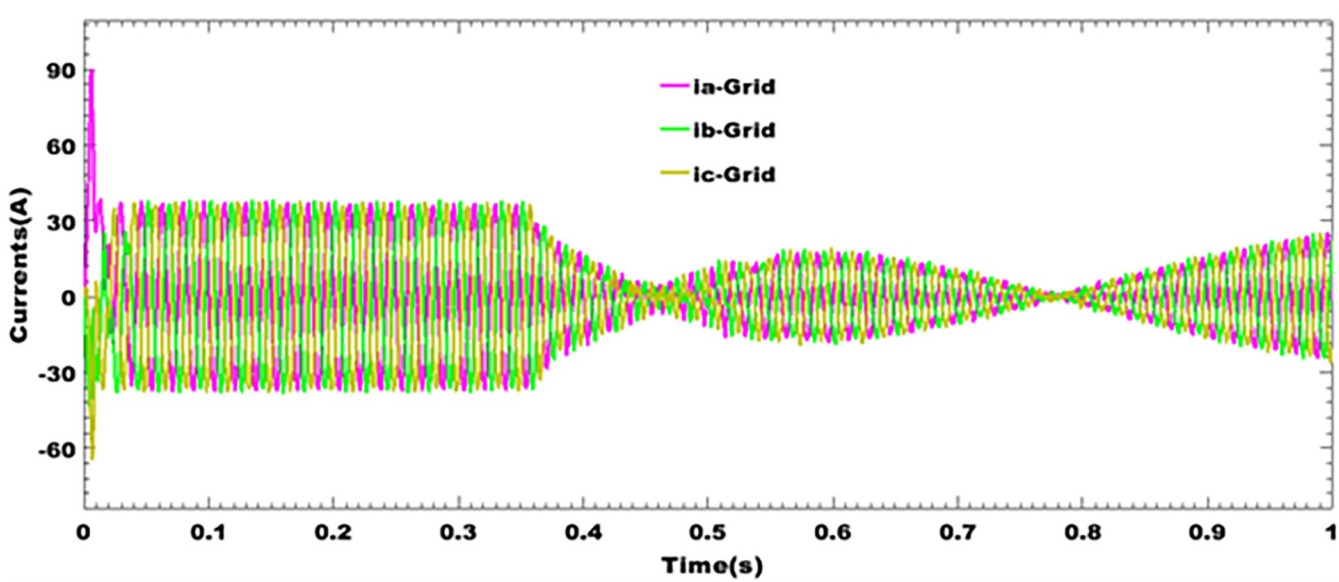

**Fig 11. Grid currents profiles during blackout.**

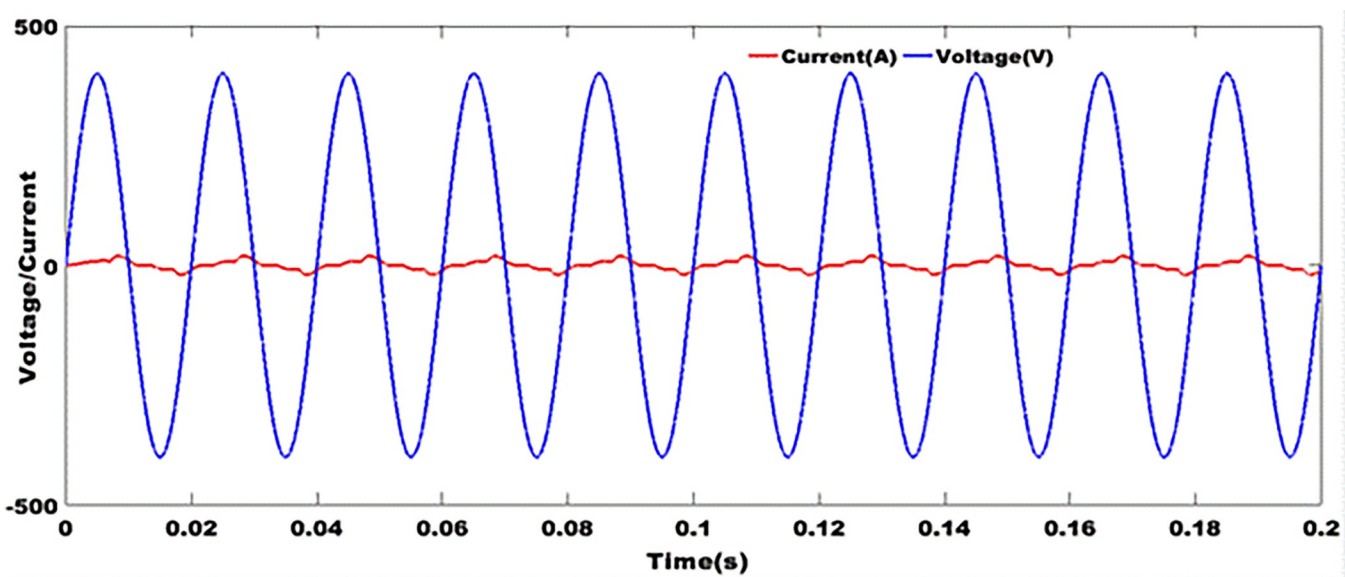

**Fig 12. Load currents and voltages after filtering.**

there is also a lack of compensation for the power absorbed. Stored energy could be used and relied on in the event of a blackout or instantaneous voltage dip. This is a fundamental reason for using battery banks to make a photovoltaic system stable and less rich in harmonics [22].

### 9.8 Total harmonics of distortion after filtering

Fig 13 reports that the harmonic distortion rate obtained at the common point of coupling is equal to 3.015% for a fundamental of 102.8, yielding a THD = 3.015%. The connection of photovoltaic sources in an electrical system is governed by the IEEE 519 standard [23], which

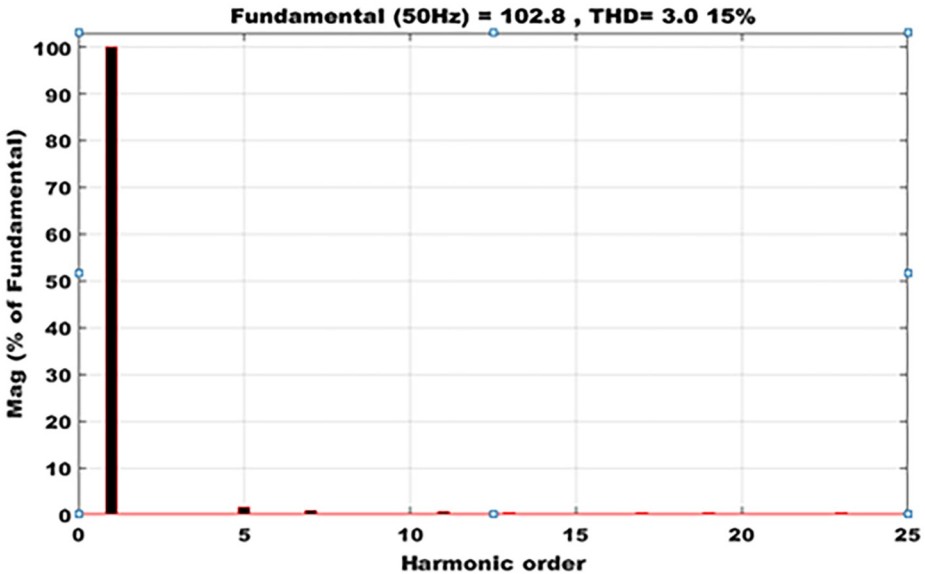

**Fig 13. Total harmonics of distortion after filtering.**

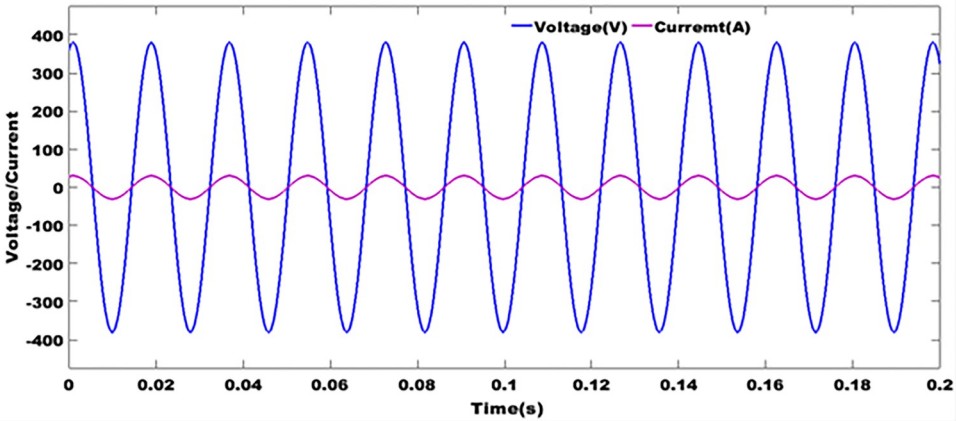

**Fig 14. Grid currents and voltages after filtering.**

imposes a rate of 5%. The rate obtained clearly shows that the proposed photovoltaic system does not inject a high rate of harmonics into the system. The odd harmonic lines disappear up to order 21. As a result, this type of decentralized generation can be fed into a centralized national grid.

### 9.9 Grid currents and voltages after filtering

From Fig 14, the grid current and voltage are in phase. In [24, 25], it is stated that when the rate of harmonic distortion is high in an electrical system, it is difficult to stabilize the network current and voltage in such a way as to keep them in phase. Since harmonic disturbances create transient fluctuations in voltage and current, they can reach peaks that are detrimental to the operation of receivers and the system as a whole. These disturbances are most often caused by non-linear loads, which send odd-rank harmonics that are multiples of the fundamental back into the network. The result shown in Fig 14 is obtained by using anti-bounce filters whose cut-off frequency is designed to reject higher frequencies. Network currents and voltages are synchronized using a phase-locked loop. This method imposes an operating frequency as a reference in a two-phase three-phase system assumed to be balanced.

### 9.10 Graph of convergence of function

Fig 15 shows the best scores obtained using the objective function, which takes as input the parameters of the whole system, whose size is to be determined. These include the number of batteries, the number of solar panels and the capacity of the battery bank. These values are listed in Table 1. The objective function provides an overall estimate of the proposed system. Through this function, the energy management system at the common point of coupling is achieved through task scheduling: injection, interruption, subscribed energy or injected energy to the grid at the PCC. Based on the work presented in [26], the PV-wind hybrid system is more robust, but unfortunately it leads to higher levels of harmonic distortion. The results obtained in Table 1 give a reasonable size for installation and maintenance operations. The proposed system is easy to maintenance and does not require any regulations other than grid connection standards [27]. The energy produced is demand-based. The size of the proposed system is suitable for integrating photovoltaic energy into an electrical system using a LCL filter, which also performs well in terms of electrical power quality [28].

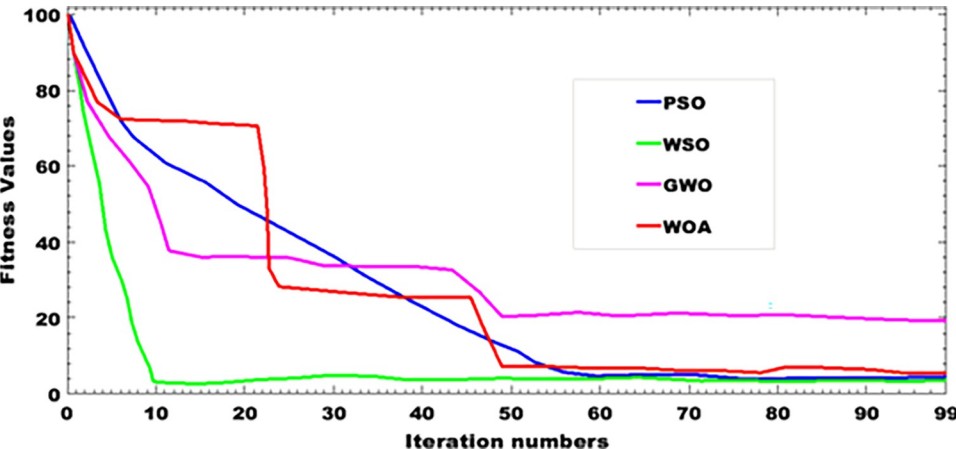

**Fig 15. Graph of convergence of PSO algorithm fitness function.**

**Table 1. Parameters.**

| Parameters | Values |
|---|---|
| $N_{PV}$ | 425 |
| $N_{Batt}$ | 112 |
| $P_{PV}$ | 8.3 MW |
| ESS | 3500 kWh |

Fig 15 depicts the fitness function defined for the different algorithms: the Grey Wolf Optimizer (GWO) [29], the White Shark Optimizer (WSO) [30], the particles swarm optimization (PSO) [31] and, the Whale Optimization Algorithm (WOA) [32]. The speed of convergence of these different algorithms makes it possible to evaluate the maximum size and power losses of the overall proposed system.

## 10. Comparison of suggested approaches

In terms of speed of convergence, the WSO algorithm converges rapidly towards the best scores compared with the other GWO, WSO and PSO algorithms. However, when looking at the power losses generated over the overall system, it can be observed that the PSO algorithm gives a lower loss than the other algorithms used in this work; power losses of 45.865 kW are observed for an active power of 453.215 kW. Whereas the other algorithms, such as the Whale Optimization Algorithm, although having an active power of 453.521, the losses are huge and approach 80.423kW. The results confirm that the PSO algorithm is better suited to the proposed method, since it converges rapidly to the best scores. Table 2 provides an analysis of the different values generated using these algorithms.

**Table 2. Comparison of different techniques.**

| Techniques | Active Power (kW) | Power losses(kW) |
|---|---|---|
| GWO [33] | 452.124 | 85.411 |
| WSO [27] | 450.850 | 86.213 |
| **PSO** [34] | **453.215** | **45.865** |
| WOA [35] | 453.521 | 80.423 |

## 11. Conclusion

The aim of this work was to evaluate a photovoltaic system connected to the grid via an LCL filter. In order to realize the different configurations of the proposed system, the boost converter was chosen along with the multilevel inverter to be associated with the LCL filter. The extraction of the maximum power point was done using the PSO whose objective function showed the best performance for evaluating not only the size of the system, but also and above all the different locations where the power generated by the system should be injected or purchased. Power quality was assessed by calculating the rate of harmonic distortion at a common point of coupling, to demonstrate the feasibility and safety of the system for all centralized generation. A rate of 3.015% was obtained in compliance with IEEE 519. This study was used not only to predict the power quality of a grid-connected photovoltaic system, but also to improve the power quality of a photovoltaic system by combining it with an electrical energy storage system, the size of which is determined by the energy demand of the household in question and that of the centralized power grid. The importance of combining battery banks has been speculated upon, given the high level of harmonic distortion in the event of the lack of a battery. The profile of the current demanded by the load is disturbed. In this unfavorable case, a current peak and a voltage dip occur, as the non-linear loads send undesirable magnitudes throughout the system. This study can be generalized to any renewable energy source. And the meta heuristic algorithms can also be used to evaluate power losses and the rate of harmonic distortion at the common point of coupling for any standard test. This method can therefore be applied to tests such as: IEEE 33 bus, IEEE 69, IEEE 118 bus.

## Supporting information

**S1 File.**
(DOCX)

## Author Contributions

**Conceptualization:** Mohammed F. Elnaggar,  Kitmo.

**Data curation:** Mohammed F. Elnaggar, Fabrice Tsegaing,  Kitmo.

**Formal analysis:** Mohammed F. Elnaggar,  Kitmo, Fabé Idrissa Barro.

**Funding acquisition:** Mohammed F. Elnaggar,  Kitmo, Fabé Idrissa Barro.

**Investigation:** Armel Duvalier Péné, Fabrice Tsegaing,  Kitmo, Fabé Idrissa Barro.

**Methodology:** Armel Duvalier Péné, André Boussaibo, Fabrice Tsegaing, Alain Foutche Tchouli,  Kitmo.

**Project administration:** Armel Duvalier Péné, André Boussaibo, Alain Foutche Tchouli, Kitmo, Fabé Idrissa Barro.

**Resources:** Armel Duvalier Péné, André Boussaibo, Alain Foutche Tchouli.

**Software:** Armel Duvalier Péné, André Boussaibo, Alain Foutche Tchouli.

**Supervision:** André Boussaibo, Fabrice Tsegaing,  Kitmo.

**Validation:**  Kitmo.

**Visualization:**  Kitmo.

**Writing – original draft:** Alain Foutche Tchouli,  Kitmo.

**Writing – review & editing:** Alain Foutche Tchouli, Kitmo.

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
