## [Decision Letter · Decision Letter 0]

5 Mar 2024

PONE-D-24-05446Optimal Sizing and Power Losses Reduction of Photovoltaic Systems using PSO and LCL FiltersPLOS ONE

Dear Dr. .,

Thank you for submitting your manuscript to PLOS ONE. After careful consideration, we feel that it has merit but does not fully meet PLOS ONE’s publication criteria as it currently stands. Therefore, we invite you to submit a revised version of the manuscript that addresses the points raised during the review process.

We look forward to receiving your revised manuscript.

Kind regards,

Vedik Basetti, Ph.D

Academic Editor

PLOS ONE

Journal Requirements:

4. Please note that funding information should not appear in any section or other areas of your manuscript. We will only publish funding information present in the Funding Statement section of the online submission form. Please remove any funding-related text from the manuscript.

5. We note that your Data Availability Statement is currently as follows: All relevant data are within the manuscript and its Supporting Information files.

Additional Editor Comments:

The manuscript proposes a method for optimizing photovoltaic systems using PSO and LCL filters, but reviewers have raised concerns about its novelty, clarity, and references. To address these, the manuscript could compare PSO with recent optimization approaches, provide convergence curves for multiple techniques, clarify references, add a flowchart for clarity, and introduce THD formulas in the introduction. Additionally, discussing limitations and future research directions would enhance the manuscript's impact and relevance in the field of photovoltaic system optimization.

Reviewers' comments:

Reviewer's Responses to Questions

**Comments to the Author**

1. Is the manuscript technically sound, and do the data support the conclusions?

Reviewer #1: Yes

Reviewer #2: Yes

2. Has the statistical analysis been performed appropriately and rigorously? 

Reviewer #1: Yes

Reviewer #2: Yes

3. Have the authors made all data underlying the findings in their manuscript fully available?

Reviewer #1: Yes

Reviewer #2: Yes

4. Is the manuscript presented in an intelligible fashion and written in standard English?

Reviewer #1: Yes

Reviewer #2: Yes

5. Review Comments to the Author

Reviewer #1: The manuscript proposes "Optimal Sizing and Power Losses Reduction of Photovoltaic Systems using PSO and LCL Filter"s.

The reviewer's concerns are as follows:

1- The novelty of this study is not clear. A flow chart showing the processes should be added to the article.

2- The formulas of THD for current and voltage should be given in the introduction part of the article.

3-Reference numbers need to be increased.

Reviewer #2: particle swarm optimization (PSO) is used to control a boost

converter and to evaluate the power losses and the harmonic distortion rate. The test

on the IEEE 14 bus standard makes it possible to determine the allocation or

integration nodes for other sources such as biomass, wind or hydrogen generators. the idea is good written but it needs further improvements like:

1. PSO is very old optimization. can you compare your results with the recent optimization approaches?

2. Convergence curve for only one optimization!. please, use three recent techniques and compared.

3. Refs. [21] & [23] are the same, please modify it.

4. Consider discussing potential limitations and future research directions to guide further exploration in this field.

5. useful citations may be used like:

10.1109/ACCESS.2023.3317980

10.1371/journal.pone.0287772

10.1016/j.asej.2022.102047

6. PLOS authors have the option to publish the peer review history of their article (what does this mean?). If published, this will include your full peer review and any attached files.

Reviewer #1: No

Reviewer #2: No

---

## [Author Response · Author response to Decision Letter 0]

13 Mar 2024

The authors would like to thank the editor and reviewers for reviewing the manuscript entitled “Optimal Sizing and Power Losses Reduction of Photovoltaic Systems using PSO and LCL Filter”, PONE-D-24-05446. The authors appreciate your insightful comments and suggestions, which have greatly contributed to improve the quality of their work. In response to your valuable feedback, the following revisions are being made in the revised manuscript. 

Note: The modifications made in the revised manuscript based on the reviewer’s comments are highlighted with yellow colour.

Response to Reviewers

Reviewer #1: 

The manuscript proposes "Optimal Sizing and Power Losses Reduction of Photovoltaic Systems using PSO and LCL Filter"s.

The reviewer's concerns are as follows:

1- The novelty of this study is not clear. A flow chart showing the processes should be added to the article.

In this paper, the particle swarm optimization (PSO) is used to control a boost converter and to evaluate the power losses and the harmonic distortion rate. The test on the IEEE 14 bus standard makes it possible to determine the allocation or integration nodes for other sources such as biomass, wind or hydrogen generators. This approach has given the possibility and feasibility to know where to inject power from other sources accordingly at each node. For instance, the THD is used to evaluate the respect of the normalized rate of harmonics at the common point of coupling.

As suggested by the respected reviewer, the flowchart of the PSO algorithms is added in the revised manuscript.

2- The formulas of THD for current and voltage should be given in the introduction part of the article.

As suggested by the respected reviewer, the THD formula is provided in equations (30 and 31) along with the revised manuscript in the introduction section.

3-Reference numbers need to be increased.

We thank the reviewer for this relevant suggestion. As suggested by the respected reviewer, reference numbers have been increased.

Reviewer #2:

 Particle swarm optimization (PSO) is used to control a boost

converter and to evaluate the power losses and the harmonic distortion rate. The test

on the IEEE 14 bus standard makes it possible to determine the allocation or

integration nodes for other sources such as biomass, wind or hydrogen generators. the idea is good written but it needs further improvements like:

1. PSO is very old optimization. can you compare your results with the recent optimization approaches?

We thank the reviewer for this relevant suggestion. As suggested by the respected reviewer, recent optimization approaches are provided to compare the obtained results. The comparison is done in table 2.

2. Convergence curve for only one optimization! please, use three recent techniques and compared.

We thank the reviewer for this relevant suggestion. As suggested by the respected reviewer, three recent techniques are used and compared with the results in table 2.

3. Refs. [21] & [23] are the same, please modify it.

We thank the reviewer for this relevant suggestion. As suggested by the respected reviewer, the duplicated reference is modified

4. Consider discussing potential limitations and future research directions to guide further exploration in this field.

We thank the reviewer for this relevant suggestion. As suggested by the respected reviewer, limitations and future research directions are given along with the manuscript.

5. useful citations may be used like:

10.1109/ACCESS.2023.3317980

10.1371/journal.pone.0287772

10.1016/j.asej.2022.102047

As suggested by the respected reviewer, these papers are cited in the revised because we found them important and suitable with our topic.

---

## [Decision Letter · Decision Letter 1]

19 Mar 2024

Optimal Sizing and Power Losses Reduction of Photovoltaic Systems using PSO and LCL Filters

PONE-D-24-05446R1

Dear Dr. .,

We’re pleased to inform you that your manuscript has been judged scientifically suitable for publication and will be formally accepted for publication once it meets all outstanding technical requirements.

Kind regards,

Vedik Basetti, Ph.D

Academic Editor

PLOS ONE

Additional Editor Comments (optional):

Reviewers' comments:

Reviewer's Responses to Questions

**Comments to the Author**

1. If the authors have adequately addressed your comments raised in a previous round of review and you feel that this manuscript is now acceptable for publication, you may indicate that here to bypass the “Comments to the Author” section, enter your conflict of interest statement in the “Confidential to Editor” section, and submit your "Accept" recommendation.

Reviewer #1: All comments have been addressed

Reviewer #2: All comments have been addressed

2. Is the manuscript technically sound, and do the data support the conclusions?

Reviewer #1: Yes

Reviewer #2: Yes

3. Has the statistical analysis been performed appropriately and rigorously? 

Reviewer #1: Yes

Reviewer #2: Yes

4. Have the authors made all data underlying the findings in their manuscript fully available?

Reviewer #1: Yes

Reviewer #2: Yes

5. Is the manuscript presented in an intelligible fashion and written in standard English?

Reviewer #1: Yes

Reviewer #2: Yes

6. Review Comments to the Author

Reviewer #1: (No Response)

Reviewer #2: All comments have been addressed. It can be accepted in the present form.

I have no competing interests.

Figures are enhanced

7. PLOS authors have the option to publish the peer review history of their article (what does this mean?). If published, this will include your full peer review and any attached files.

Reviewer #1: **Yes: **Asst.Professor Suleyman Adak

Reviewer #2: No

---

## [Editor Report · Acceptance letter]

24 Mar 2024

PONE-D-24-05446R1 

PLOS ONE

Dear Dr. ., 

I'm pleased to inform you that your manuscript has been deemed suitable for publication in PLOS ONE. Congratulations! Your manuscript is now being handed over to our production team.

Kind regards, 

on behalf of

Dr. Vedik Basetti 

Academic Editor

PLOS ONE